# Curability difference between autochthonous mouse tumors and their transplants in association with immune gene expression

Hiroshi Tanooka[ID][1]*, Chie Kudo-Saito[2], Fumiko Chiwaki[3], Masamichi Ishiai[1], Kouichi Tatsumi[4‡], Hiroki Sasaki[3†], Takahiro Ochiya[5]

1 Central Radioisotope Division, National Cancer Center Research Institute, Tokyo, Japan, 2 Department of Immune Medicine, National Cancer Center Research Institute, Tokyo, Japan, 3 Department of Translational Oncology, National Cancer Center Research Institute, Tokyo, Japan, 4 Biology Division, National Institute of Radiological Sciences, Chiba, Japan, 5 Department of Molecular and Cellular Medicine, Institute of Medical Science, Tokyo Medical University, Tokyo, Japan

† Deceased in May 2025.
‡ Retired.
* tanooka-h@wind.ocn.ne.jp

## Abstract

Compared with transplanted tumors, autochthonous tumors are difficult to cure using experimental radiation therapy in mice. Here we analyzed differences in immune-related gene expression profiles between mouse fibrosarcomas subcutaneously induced by 3-methylcholanthrene (3MC) and their corresponding transplanted tumors. The immune genes examined were *Pd1*, *Pdl1*, *Pdl2*, *Cd3d*, *Cd8a*, *Cd8b*, *Ifnγ*, *Itga2*, *Gzmb,* and *Foxp3*. Among 12 tumors, one was non-transplantable and showed a benign phenotype with an abundance of DX5$^+$ natural killer cells and CD8$^+$ T cells together with increased IFNγ expression and mRNA levels of all immune genes except for *Itga2*. The other 11 transplantable tumors showed increased expression of *Pd1*, *Pdl1*, *Pdl2*, *Cd3d*, *Cd8b*, and *Ifnγ* following transplantation into syngeneic mice. These effects of transplantation highlight the relevance of immune gene expression status to the curability of tumors.

## Introduction

Transplanted mouse tumors are widely applied as a model tumor system to experimental radiation therapy [1], while application of autochthonous tumors, such as spontaneous mammary carcinoma [2], is very rare. Autochthonous tumors, compared with transplanted tumors, are difficult to cure by experimental therapy, as noted by Nakahara [3]. According to the strict definition of cure as complete eradication of a tumor mass followed by mouse survival for 120 days without recurrence, fibrosarcomas induced with 3MC in C57BL/6J mice are incurable after experimental radiation therapy with a collimated X-ray beam, with few exceptions, whereas tumors

**Data availability statement:** All relevant data are within the paper and its Supporting information files.

**Funding:** This work was supported by a Japan Agency for Medical Research and Development Grant (18ae0101011h005) and a Project for Cancer Research and Therapeutic Evolution Grant (JP20cm0106402) to TO and a National Cancer Center Research and Development Fund (2023-J-02) to MI.

**Competing interests:** The authors have declared that no competing interests exist.

transplanted in syngeneic mice are completely curable in a radiation dose-dependent manner [4] (S1 Fig).

The reason for this distinct difference has long remained unclear. One explanation is that there is new tumor formation at the 3MC-treated site in autochthonous tumors, as it is exposed to the chemical carcinogen 3MC and later to radiation. The recurrent tumors, however, are clones of the original tumors, as demonstrated from analysis of isozyme patterns of tumors in *Pgk-1/Pgk-2* mosaic cell mice [5], indicating true recurrence. One other possible explanation for the curability difference is a change in the immune status of the tumor microenvironment organismally after transplantation.

Recent findings show that radiation not only directly kills cancer cells, but also induces expression of anti-cancer immune genes [6,7] and the immuno-transmitter [8], and further that the immune activation is more efficient with high LET than low LET radiation [9,10]. This radiation effect might be related to the transplantation effect for tumor cure.

Here we focused on the expression of immune genes in 3MC-induced mouse tumors before and after transplantation. Target genes were programmed cell death protein (*Pd1*) [11], its blockade ligands *Pdl1* [12] and *Pdl2* [13], the T-cell marker *Cd3d* [14], cytotoxic T-cell markers *Cd8a* [15] and *Cd8b* [15], interferon gamma (*Ifnγ*) [16], integrin *Itga2* [17], granzyme *Gzmb* [18], and the regulatory T-cell marker *Foxp3* [19].

## Materials and methods

### Mice

The experiments were approved by the National Cancer Center (T21-006) and carried out in consideration of the Guideline for Animal Experiments in the National Cancer Center and the 3Rs stipulated by the Animal Welfare Management Act. Specific pathogen-free (SPF) female C57BL/6J mice at 10 weeks of age were purchased from Charles River Japan and kept in an isolating rack with four mice at maximum in the SPF animal room with an animal diet and water *ad libitum*.

### Tumor induction and transplantation

An aliquot of 0.1 ml of 3MC (Sigma) dissolved in 5 mg/ml in olive oil was injected at the groin of mice as described previously [4]. Mice were observed for tumor formation by palpation. When tumors grew to the size of 1 cm in diameter, they were resected and divided into sections for histological examination, transplantation, and gene expression measurements (Fig 1A). A piece of tumor was transplanted subcutaneously into the groin of a female C57BL/6J mouse with a transplantation needle (13 G) in duplicate. Expanded transplants were examined in the same way as for autochthonous tumors. Liver and lung specimens were obtained at the same time as tumor resection. Tumors and organs were kept frozen at −80° C.

### Histology

Tumor sections were fixed in 10% formalin, embedded in paraffin, sliced, stained with hematoxylin and eosin, and examined histologically under a microscope.

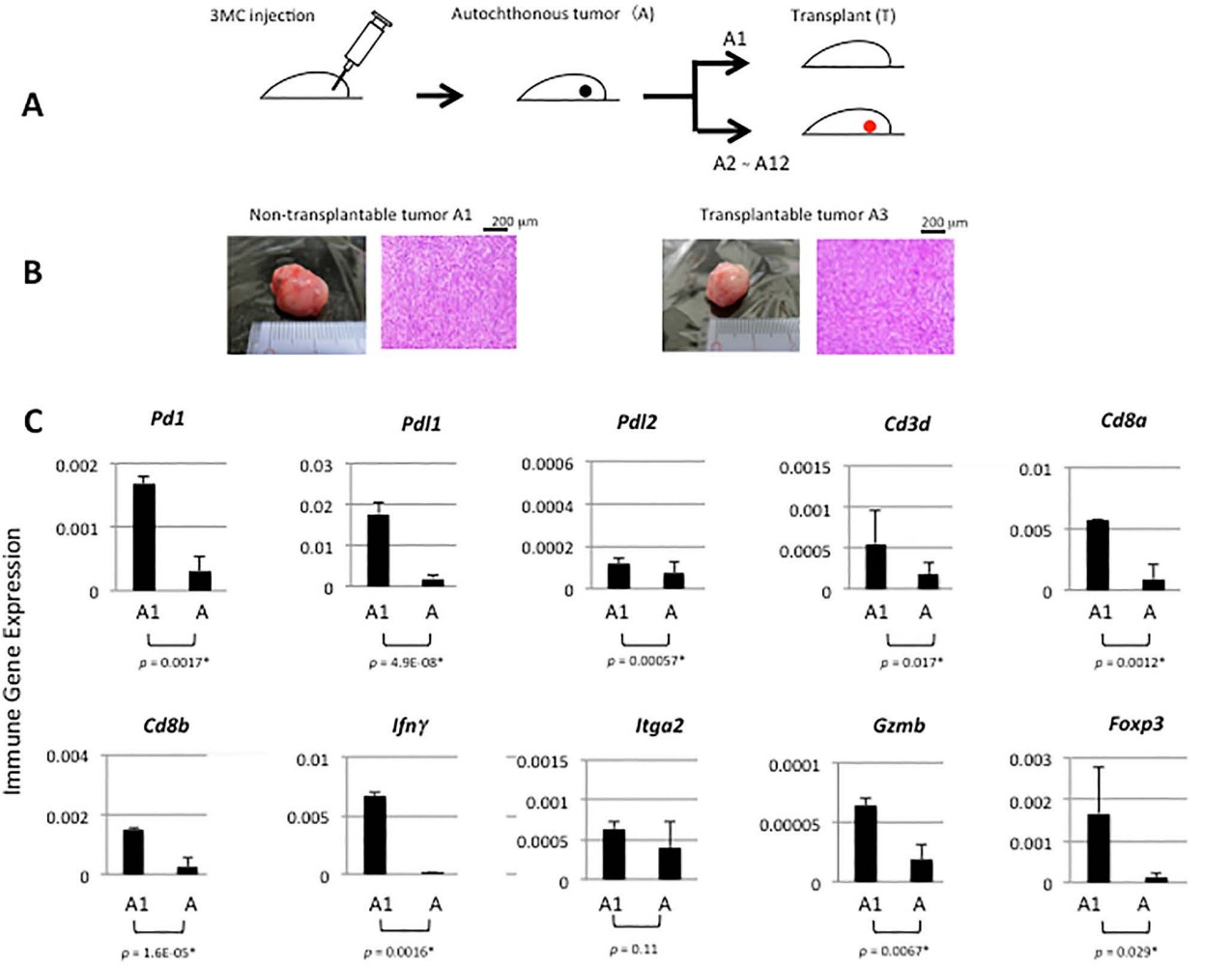

**Fig 1. Protocol of experiments, histology and immune gene expression of non-transplantable and transplantable tumors. A.** Tumor induction with 3MC in female C57BL/6j mice and transplantation to syngeneic mice. **B.** Non-transplantable tumor, A1, compared with one of transplantable tumors, A3, showing the indistinguishable histological type of fibrosarcoma of the skin. **C.** Expressions of mRNAs of immune genes in the A1 tumor, measured by qPCR, were compared with the average level of 11 transplantable tumors, A (*: statistically significant, p < 0.05).

### Immune gene expression measurement with qPCR

Two sections of each tumor were homogenized in a lysis reagent (QIAzol, Qiagen) with a homogenizer (Kinematica), and total RNA was extracted with an extraction kit (miRNeasy, Qiagen) and converted to cDNA with a reverse transcription kit (High Capacity cDNA Transcription kit, Applied Biosystems), according to the manufacturer's protocol. qPCR was performed at 95 °C for 3 min followed by 40 cycles at 95 °C for 10 s and at 55 °C for 30 s with the Real-Time qPCR System (CFX90, Bio-Rad), using a reaction mixture (TaqMan Fast Advanced Master Mix, Applied Biosystems) with probes for measurements of immune gene expression, according to the manufacturer's protocol (TaqMan Gene Expression Assays): *Pd1* (Mm01285676_ml), *Pdl1* (Mm03043248_ml), *Pdl2* (Mm00451734_ml), *Cd3d* (Mm00442746_ml), *Cd8a* (Mm01182107_gl), *Cd8b* (Mm00438116_ml), *Ifnγ*(Mm01168134_ml), *Itga2* (Mm00434371_ml), *Gzmb* (Mm00493152_gl), *Foxp3* (Mm0047516_ml), and *β-actin* (Mm00607939_sl) as a control. Expression levels for immune genes were analyzed using Excel and calibrated to those for *β-actin*.

                                                                                    

## Immunohistochemical analysis

The paraffin-embedded tumor sections were stained with the following antibodies according to standard methods [20]: anti-DX5-FITC (BioLegend #108909), anti-CD8-PE-Cy5 (BD Bioscience #553034), anti-IFNγ-PE (BD Bioscience #554412), and the appropriate isotype controls (BD Bioscience). Three locations per section were observed at 100× magnification using a LSM700 laser scanning confocal microscope (Carl Zeiss), and the number of DX5+ NK cells and CD8+ T cells contained in the field was counted. The immunofluorescence intensity of IFNγ expression was automatically measured as pixel counts at three locations per section using the ZEN 2012 software installed in the LSM700 microscope. The average of the data was used in graphs.

## Statistics

Student's t-test was applied to statistical analysis of measured data, using Excel program.

## Results and discussion

Autochthonous tumors induced subcutaneously with 3MC in female C57BL/6J mice were compared to transplants for immune gene expression in the context of differences in curability after experimental radiation therapy. Among 13 mice, 12 animals produced tumors (designated as A1–A12). The first tumor was detected 76 days after MC injection. All were fibrosarcomas of the skin and histologically indistinguishable and were transplanted into syngeneic female mice (Fig 1A).

### High infiltration of anti-tumor effector cells in a non-transplantable tumor

Among 12 tumors, one tumor (A1) failed to engraft after four attempts. The other 11 tumors (A2–A12) were transplant-able. The A1 tumor was not visibly different from the other transplantable tumors (see Fig 1B for an example comparison). After qPCR analysis of mRNA/cDNA, A1 showed the unique feature of highly expressing *Pd1*, *Pdl1*, *Pdl2*, *Cd3d, Cd8a*, *Cd8b*, *Ifnγ*, *Gzmb*, and *Foxp3* as compared to transplantable tumors (Fig 1C). Indeed, immunohistochemical analysis comparing A1 with transplantable tumor tissues highlighted a significantly greater abundance of DX5+ natural killer (NK) cells (Fig 2A) and CD3+CD8+ T cells (Fig 2B) and a significantly higher expression of IFNγ in the infiltrating CD8+ T cells (Fig 2C) in the A1 tumor. Of note, a few CD8+ T cells but no NK cells were observed within the transplant-able tumors, and NK cells in the A1 tumor were much larger than the CD8+ T cells (Fig 2D), suggesting an activated state. These results indicate that these anti-tumor effector cells, particularly NK cells, were involved in the benign character of the A1 tumor.

In a previous study, a few 3MC-induced autochthonous tumors (3 of 28; 11%) were curable after experimental radiation therapy [4]. This frequency largely coincides with non-transplantable tumor frequency in the current study (1 of 12; 8%), indicating that benign tumors can be present in a histologically indistinguishable tumor population even when induced in the same syngeneic mouse strain by the same method.

In the liver of the A1-bearing host mouse, immune gene expression levels did not differ from that in livers of mice bearing transplantable tumors or livers of untreated control mice, while the lung of A1-bearing mouse showed a higher expression of *Pd1*, *Pdl1*, *Pdl2*, *Ifnγ*, *Itga2*, compared with the lung of mice carrying transplantable tumors (S2 Fig). How-ever, this level was at the same level as in untreated control mice, suggesting the possibility that the immune response is suppressed in the 3MC-treated mice carrying transplantable tumors, as shown in the early work [21].

### Immune gene expression in autochthonous tumors and transplants

3MC-induced autochthonous tumors were monitored to access immune gene expression with their transplantation into syngeneic mice. The expression levels of *Pd1*, *Pdl1*, *Pdl2*, *Cd3d*, *Cd8b, and Ifnγ* were significantly increased with trans-plantation (Fig 3).

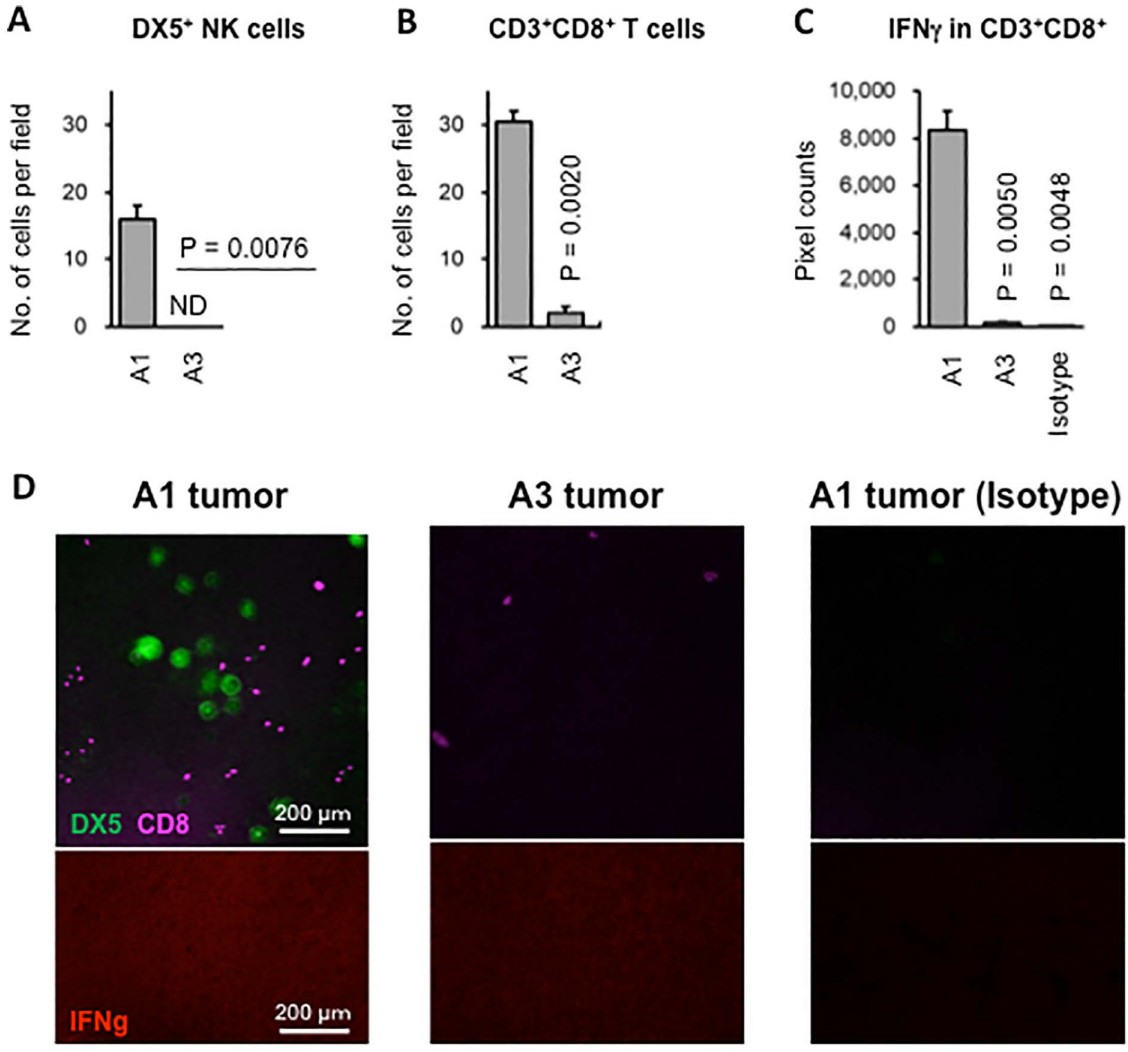

**Fig 2. High infiltration of anti-tumor effector cells in non-transplantable tumor.** Sections of non-transplantable tumor A1and transplantable tumor A3 were stained with anti-DX5-FITC, anti-CD8-PE-Cy5, anti-IFNγ-PE, and the appropriate isotype controls. **A.** Infiltration of DX5+ NK cells. ND, not detected. **B.** Infiltration of CD8+ T cells. **C.** Intensity of IFNγ expression. **D.** Representative photos of A1 and A3 tumors after immuno-histochemical staining.

The co-expression of these genes may appear paradoxical, as they include the inhibitory receptor *Pd1*, its ligands *Pdl1* and *Pdl2*, and T-cell markers *Cd3d* and *Cd8b* associated with anti-tumor immunity. However, the concurrent expression of *Pd1, Pdl1*, and *Pdl2* is not contradictory but rather reflects coordinated activation of immune checkpoint pathways. Such co-expression has been widely reported in cancer and is associated with inflammatory signaling, adaptive immune resistance, and T-cell exhaustion [22,23]. Notably, co-expression of PD1 and PDL1 has been reported to correlate with favorable postoperative prognosis in human gastric cancer [24]. Furthermore, inflammatory IFNγfeeds back to upregulate expression of immune checkpoint genes [25]. The results of the present study are consistent with these findings.

The variability of measured values in Fig 3 likely reflects the inhomogeneous clonal composition of autochthonous tumors. Nevertheless, the malignant characteristics and the immune activation appear to be common features of transplantable autochthonous tumors. Cell death induced by transplantation or irradiation may generate similar signals within

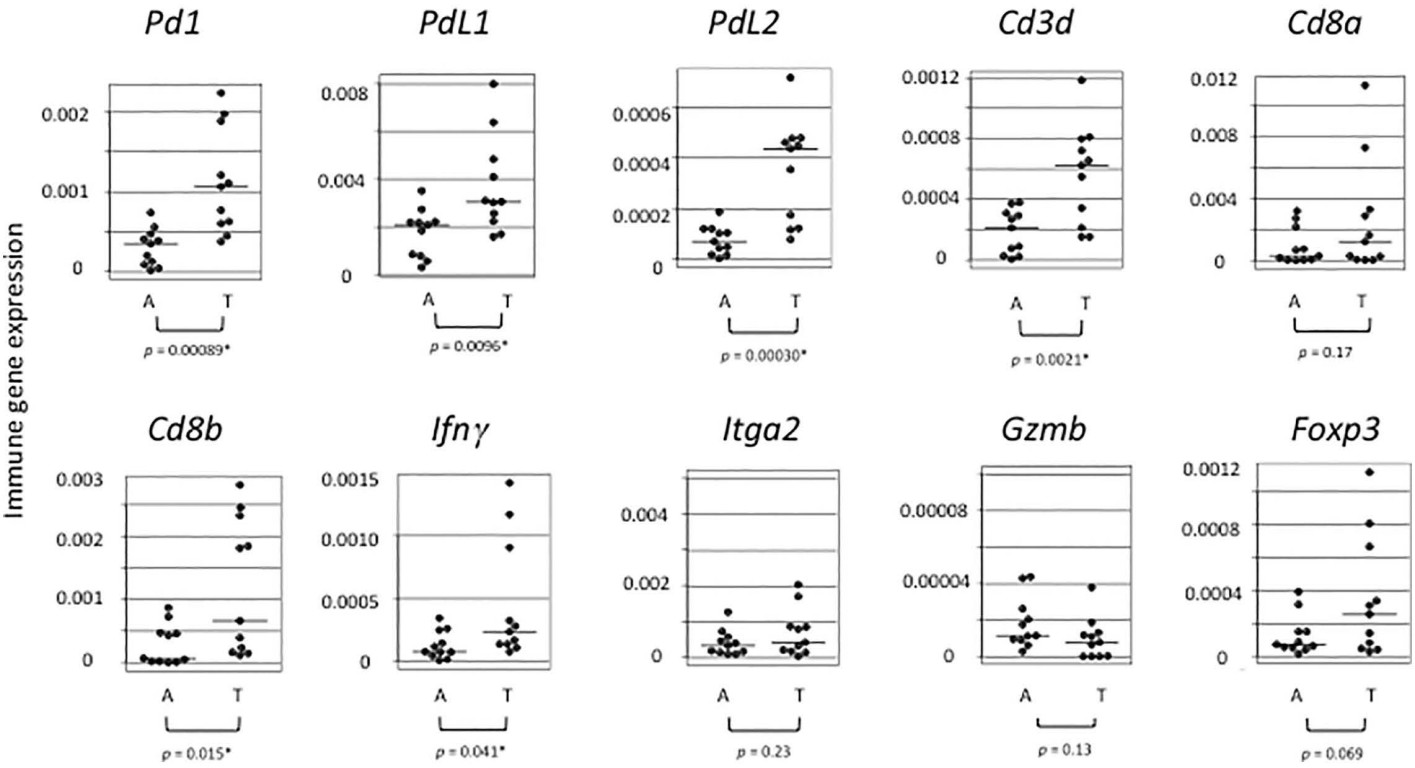

**Fig 3. Immune gene mRNA expression in autochthonous tumors and their transplants measured by qPCR.** Transplantable 3MC-induced autochthonous fibrosarcomas, A (n = 11), were compared with their transplants, T (n = 11), for immune gene expression. Average expression levels are shown by horizontal bars. (*: statistically significant, *p* < 0.05).

the tumor microenvironment, thereby promoting cytotoxic T-cell infiltration. In the present study, the benign A1 tumor may also generate comparable signals. Whether radiation-induced immune responses differ between primary autochthonous tumors and their transplanted counterparts remains an important question for future investigation.

Additional evidence presents a complex picture. Early studies of the immunogenic nature of 3MC-induced mouse tumors indicated that the host mouse had sufficient immune activity to reject the replanted tumor, whereas untreated mice did not [26,27], indicating a high anti-tumor immune activity in host animals. Although curability and the immune gene expression profile of tumors were not described in these studies, the findings indicate that host mice bearing autochthonous tumors have a high immunogenic activity and more curable tumors compared with transplanted tumors.

However, this interpretation contrasts with later reports describing incurable autochthonous tumors [4].

## Conclusion

Immune gene expression in 3MC-induced autochthonous tumors was significantly increased following transplantation. This enhancement is consistent with the observed differences in curability between primary autochthonous tumors and their transplants. Furthermore, the benign, non-transplantable phenotype strongly correlated with elevated immune gene expression and increased infiltration of anti-tumor effector cells.

## Supporting information

**S1 Fig. Curability difference between 3MC-induced autochthonous tumors and their transplants.**
(DOCX)

**S2 Fig. Immune gene expressions in the mouse liver and lung as measured by qPCR.**
(DOCX)

**S1 File. Protocols.**
(DOCX)

**S1 Data. Data deposit.**
(XLSX)

## Acknowledgments

We thank the late W. Nahakahara and the late R. Tokuzen for advice regarding the 3MC injection method; T. Sado, K. Shimotohno, and H. Nishikawa for useful suggestions and discussions; R. Machinami and A. Yoshida for histological examination; Y. Yamamoto for help in constructing figures; N. Uchiya and Y. Shiotani for preparing histological sections; and the animal facility staff at the National Cancer Center for their expert animal care.

## Author contributions

**Conceptualization:** Hiroshi Tanooka, Hiroki Sasaki, Takahiro Ochiya.

**Data curation:** Hiroshi Tanooka, Chie Kudo-Saito, Fumiko Chiwaki.

**Formal analysis:** Hiroshi Tanooka, Chie Kudo-Saito.

**Funding acquisition:** Masamichi Ishiai, Takahiro Ochiya.

**Investigation:** Chie Kudo-Saito, Fumiko Chiwaki, Masamichi Ishiai, Kouichi Tatsumi, Hiroki Sasaki, Takahiro Ochiya.

**Methodology:** Hiroshi Tanooka, Chie Kudo-Saito, Fumiko Chiwaki, Hiroki Sasaki.

**Project administration:** Masamichi Ishiai, Takahiro Ochiya.

**Supervision:** Kouichi Tatsumi, Takahiro Ochiya.

**Validation:** Hiroshi Tanooka.

**Visualization:** Hiroshi Tanooka.

**Writing – original draft:** Hiroshi Tanooka, Chie Kudo-Saito.

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
