## [Decision Letter · Decision Letter 0]

13 Feb 2026

Dear Dr. Tanooka,

Thank you for submitting your manuscript to PLOS ONE. After careful consideration, we feel that it has merit but does not fully meet PLOS ONE’s publication criteria as it currently stands. Therefore, we invite you to submit a revised version of the manuscript that addresses the points raised during the review process.

Please respond to reviewers' comments individually.

We look forward to receiving your revised manuscript.

Kind regards,

Xiaosheng Tan

Academic Editor

PLOS One

Journal Requirements:

https://journals.plos.org/plosone/s/file?id=ba62/PLOSOne_formatting_sample_title_authors_affiliations.pdf....

“This work was supported by a Japan Agency for Medical Research and Development Grant (18ae0101011h005) and a Project for Cancer Research and Therapeutic Evolution Grant (JP20cm0106402) to TO and a National Cancer Center Research and Development Fund (2023-J-02) to MI.”

“This work was supported by a Japan Agency for Medical Research and Development Grant (18ae0101011h005) and a Project for Cancer Research and Therapeutic Evolution Grant (JP20cm0106402) to TO and a National Cancer Center Research and Development Fund (2023-J-02) to MI.”

5. In this instance it seems there may be acceptable restrictions in place that prevent the public sharing of your minimal data. However, in line with our goal of ensuring long-term data availability to all interested researchers, PLOS’ Data Policy states that authors cannot be the sole named individuals responsible for ensuring data access (http://journals.plos.org/plosone/s/data-availability#loc-acceptable-data-sharing-methods).

6. Please be informed that funding information should not appear in the Acknowledgments section or other areas of your manuscript. We will only publish funding information present in the Funding Statement section of the online submission form. Please remove any funding-related text from the manuscript.

7. Please include a separate caption for each figure in your manuscript.

Reviewers' comments:

Reviewer's Responses to Questions

**Comments to the Author**

1. Is the manuscript technically sound, and do the data support the conclusions?

Reviewer #1: Partly

Reviewer #2: Partly

2. Has the statistical analysis been performed appropriately and rigorously?

Reviewer #1: Yes

Reviewer #2: Yes

3. Have the authors made all data underlying the findings in their manuscript fully available?

Reviewer #1: Yes

Reviewer #2: Yes

4. Is the manuscript presented in an intelligible fashion and written in standard English?

Reviewer #1: Yes

Reviewer #2: Yes

Reviewer #1: The study compared the expression differences of immune-related genes in spontaneously induced 3MC-induced fibrosarcoma with homologous xenograft tumors, proposing that "upregulation of immune gene expression after transplantation may be associated with changes in curability."

1. The text directly interprets "upregulation of immune genes after transplantation" as "at least partially explaining curability change," but this study did not conduct a new radiotherapy cure experiment, relying primarily on previous studies and the S1 Fig background.

The conclusion should be worded differently, changing "explain" to "is consistent with / may contribute to / is associated with," and explicitly stating that "this study is an observation of the correlation between expression profiles and immune infiltration." If the authors wish to retain the strong conclusion of "explaining differences in curability," they should supplement it with at least one set (even small-scale) of immune gene/infiltration changes before and after transplantation plus radiotherapy, or functional validation (such as CD8/NK depletion, PD-1/PD-L1 blockade, etc.) to connect "expression changes to curability."

2. There are obvious errors in the terminology/annotations of key immune markers, which will affect credibility. DX5 is not "perforin," but a commonly used NK cell surface marker (CD49b). grzmb is granzyme B, not integrin (and the spelling should be consistent: grzmb/gzmb has been used interchangeably in the text). It is recommended to clarify in the methods and figures whether the qPCR is measuring Itga2/CD49b or another gene. Currently, the gene corresponding to "dx5 (Mm00434371_ml)" needs to be verified and its official gene name written (avoid using antibody names as gene names).

"datys" is misspelled. "pll2" is suspected to be a misspelling of pdl2. It is recommended to standardize the naming of immune genes as "PD-1, PD-L1, PD-L2, Cd3d…" or follow mouse gene naming conventions (capitalized first letter, lowercase the rest), and maintain consistency throughout the text (the current interchangeable use of pd1/pdL1, etc., in the abstract/main text/figures).

3. The descriptions of the animals and their husbandry are contradictory and need to be more standardized. The method is described as "Germ-free mice" but it also says "kept in SPF room".

Reviewer #2: This is a concise, focused study examining immune gene expression differences between autochthonous 3MC-induced mouse fibrosarcomas and their transplants, with the aim of explaining the well-known but poorly understood phenomenon of superior transplant curability by radiation therapy. The study addresses a genuine and long-standing question in experimental cancer biology. However, the manuscript is quite limited in scope and the conclusions drawn exceed what the data can support.

Strength: The central research question is biologically meaningful and historically relevant, addressing a phenomenon first noted decades ago. The experimental design is straightforward and the use of matched autochthonous/transplant pairs from the same tumors is a conceptual strength. The identification of a non-transplantable, apparently "benign" tumor (A1) with a dramatically distinct immune profile is the most interesting finding in the paper and is convincingly supported by both qPCR and immunohistochemical data. The figures are clear and the statistical comparisons are appropriately applied.

Concern: The central argument — that increased immune gene expression after transplantation explains the curability difference — is not established by the data presented. The study shows that certain immune genes (pd1, pdl1, pdl2, cd3d, cd8b) are upregulated in transplants compared to autochthonous tumors, but it provides no evidence that this change is causally related to improved curability. Paradoxically, several of the upregulated genes (pd1, pdl1, pdl2) are canonical immunosuppressive checkpoint markers, not anti-tumor effectors. Upregulation of PD-1/PD-L1/PD-L2 would classically be interpreted as increased immune suppression, yet the authors frame this as evidence of enhanced immune activity that facilitates tumor cure. This central interpretational contradiction is never addressed and significantly undermines the manuscript's main conclusion.

.

Reviewer #1: No

Reviewer #2: No

---

## [Author Response · Author response to Decision Letter 1]

5 Mar 2026

Response to Reviewer #1

1. Comment:

Reviewer #1 pointed out that the statement “upregulation of immune genes after transplantation as at least partially explaining the curability change” is too strong.

Response:

We appreciate the reviewer’s comment and have revised the statement accordingly. In the Conclusion (page 7), we now state:

“This increase is consistent with the difference in curability between autochthonous tumors and their transplants.”

2. Comment:

The reviewer pointed out errors in terminology.

Response:

We carefully reviewed and corrected the terminology according to Mouse Genome Informatics (MGI) guidelines throughout the text, figures, and supplementary figures (all corrections are marked in red). Mouse gene names are now formatted with an initial capital letter followed by lowercase letters and italicized throughout.

As noted by the reviewer, the mouse gene corresponding to the DX5 protein (an NK cell surface marker) is Itgb2 (CD49b). Therefore, “Dx5” has been corrected to Itgab. The qPCR TaqMan probe (Mm00434371_m1) used in this study targets mouse Itgab.

The gene name for granzyme B has been corrected to Gzmb.

Misspellings (“datys” and “pll2”) have also been corrected.

3. Comment:

The reviewer questioned the description of the animals.

Response:

We thank the reviewer for pointing this out. The term “germ-free mice” was used inadvertently. The mice used in this study were specific pathogen-free (SPF), not germ-free. We have corrected “germ-free mice” to “specific pathogen-free (SPF) mice” (page 4, Mice in Materials and Methods).

Response to Reviewer #2

1. Comment:

The reviewer noted that the central argument- “increased immune gene expression after transplantation explains the curability difference”- is not established by the presented data. The co-expression of the immune effector PD1 and the immunosuppressive molecules PDL1 and PDL2 also appears paradoxical.

Response:

We agree that our data do not provide direct evidence that upregulation of immune genes after tumor transplantation causally explains the increase in curability. We have therefore revised the manuscript to avoid overstatement.

Although we cannot demonstrate causality, the upregulation of immune-related genes may reflect inducible immune activation associated with host defense responses.

To address this point, we have added the following discussion (pages 6–7):

“The co-expression of these genes may appear paradoxical, as they include the inhibitory receptor PD1, its ligands PDL1 and PDL2, and T-cell markers Cd3d and Cd8b associated with anti-tumor immunity. However, this pattern likely reflects coordinated induction by common upstream immune-activating signals, although the precise mechanisms remain unclear. Notably, co-expression of PD1 and PDL1 has been reported to correlate with favorable postoperative prognosis in human gastric cancer [22].”

All textual revisions, including corrections in figures and supplementary figures, are marked in red. Sentences modified for clarity and English expression are marked in blue (page 7).

---

## [Decision Letter · Decision Letter 1]

20 Mar 2026

Dear Dr. Tanooka,

Thank you for submitting your manuscript to PLOS ONE. After careful consideration, we feel that it has merit but does not fully meet PLOS ONE’s publication criteria as it currently stands. Therefore, we invite you to submit a revised version of the manuscript that addresses the points raised during the review process.

Please respond to reviewer's comments.

We look forward to receiving your revised manuscript.

Kind regards,

Xiaosheng Tan

Academic Editor

PLOS One

Journal Requirements:

Reviewers' comments:

Reviewer's Responses to Questions

**Comments to the Author**

Reviewer #1: All comments have been addressed

Reviewer #2: (No Response)

2. Is the manuscript technically sound, and do the data support the conclusions?

Reviewer #1: Yes

Reviewer #2: Yes

3. Has the statistical analysis been performed appropriately and rigorously?

Reviewer #1: Yes

Reviewer #2: Yes

4. Have the authors made all data underlying the findings in their manuscript fully available?

Reviewer #1: Yes

Reviewer #2: Yes

5. Is the manuscript presented in an intelligible fashion and written in standard English?

Reviewer #1: Yes

Reviewer #2: Yes

Reviewer #1: (No Response)

Reviewer #2: The authors have made genuine efforts to address the concerns raised in the first round of review, and the revised manuscript is more measured in its claims. The terminology corrections, the SPF mouse clarification, and the softened causal language in the Conclusion are all appropriate improvements.

However, the interpretational issue regarding the upregulation of Pd1, Pdl1, and Pdl2 in transplanted tumors has not been adequately resolved. These three genes are canonical checkpoint inhibitory markers whose upregulation is classically associated with immune suppression, yet the manuscript continues to frame their increased expression as broadly consistent with enhanced anti-tumor immunity and improved curability.

.

Reviewer #1: **Yes:**Chen FengChen FengChen FengChen Feng

Reviewer #2: No

---

## [Author Response · Author response to Decision Letter 2]

31 Mar 2026

Second Response to Reviewers’ Comments (March 29, 2026)

First responses to reviewers comments were described in the Response of Februray 24, 2026.

Reviewer #1

Reviewer #1 had no comment.

Reviewer #2

Reviewer #2 raised a further question concerning the simultaneous upregulation of Pd1, Pdl1, and Pdl2.

We respond to this question as follows.

“ The co-expression of these genes may appear paradoxical, as they include the inhibitory receptor Pd1, its ligands Pdl1 and Pdl2, and T-cell markers Cd3d and Cd8b associated with anti-tumor immunity. However, the concurrent expression of Pd1, Pdl1, and Pdl2 is not contradictory but rather reflects coordinated activation of immune checkpoint pathways. Such co-expression has been widely reported in cancer and is associated with inflammatory signaling, adaptive immune resistance, and T-cell exhaustion [22, 23]. Notably, co-expression of PD1 and PDL1 has been reported to correlate with favorable postoperative prognosis in human gastric cancer [24]. Furthermore, inflammatory IFNg feeds back to upregulate expression of immune checkpoint genes [25]. The results of the present study are consistent with these findings. (from page 6, line 168 to page 7, line 177)

---

## [Editor Report · Decision Letter 2]

7 Apr 2026

Curability Difference between Autochthonous Mouse Tumors and Their Transplants in Association with Immune Gene Expression

PONE-D-25-62358R2

Dear Dr. Tanooka,

We’re pleased to inform you that your manuscript has been judged scientifically suitable for publication and will be formally accepted for publication once it meets all outstanding technical requirements.

Kind regards,

Xiaosheng Tan

Academic Editor

PLOS One
---

## [Editor Report · Acceptance letter]

PONE-D-25-62358R2

PLOS One

Dear Dr. Tanooka,

I'm pleased to inform you that your manuscript has been deemed suitable for publication in PLOS One. Congratulations! Your manuscript is now being handed over to our production team.

Kind regards,

on behalf of

Dr. Xiaosheng Tan

Academic Editor

PLOS One